# Mitochondrial network structure controls cell-to-cell mtDNA variability generated by cell divisions

**Robert C. Glastad**[1], **Iain G. Johnston**[1,2]*

**1** Department of Mathematics, University of Bergen, Bergen, Norway, **2** Computational Biology Unit, University of Bergen, Bergen, Norway

* iain.johnston@uib.no

## Abstract

Mitochondria are highly dynamic organelles, containing vital populations of mitochondrial DNA (mtDNA) distributed throughout the cell. Mitochondria form diverse physical structures in different cells, from cell-wide reticulated networks to fragmented individual organelles. These physical structures are known to influence the genetic makeup of mtDNA populations between cell divisions, but their influence on the inheritance of mtDNA at divisions remains less understood. Here, we use statistical and computational models of mtDNA content inside and outside the reticulated network to quantify how mitochondrial network structure can control the variances of inherited mtDNA copy number and mutant load. We assess the use of moment-based approximations to describe heteroplasmy variance and identify several cases where such an approach has shortcomings. We show that biased inclusion of one mtDNA type in the network can substantially increase heteroplasmy variance (acting as a genetic bottleneck), and controlled distribution of network mass and mtDNA through the cell can conversely reduce heteroplasmy variance below a binomial inheritance picture. Network structure also allows the generation of heteroplasmy variance while controlling copy number inheritance to sub-binomial levels, reconciling several observations from the experimental literature. Overall, different network structures and mtDNA arrangements within them can control the variances of key variables to suit a palette of different inheritance priorities.

**Data Availability Statement:** All code is freely and publically available at https://github.com/StochasticBiology/mtdna-network-partition.

**Funding:** This project has received funding from the European Research Council (ERC) under the

## Author summary

In many organisms, mitochondria form large, connected networks. The reasons for this network formation are not fully understood, and it is likely that several different advantages may be provided by a physical network structure. Here we use maths and simulation to show and explore one of these possible advantages. By forming physical networks in the cell, mitochondria can control the inheritance of the vital mtDNA molecules that they contain. Different physical behaviour of mitochondrial networks can both generate useful variability in, and tightly control, the genetic mtDNA content of daughter cells after divisions. This physical control of genetic content allows different priorities to be addressed,

European Union's Horizon 2020 research and innovation programme (Grant agreement No. 805046 (EvoConBiO) to IGJ). The funders had no role in study design, data collection and analysis, decision to publish, or preparation of the manuscript.

**Competing interests:** The authors have declared that no competing interests exist.

including the segregation of mutational damage, and the faithful inheritance of mtDNA copy number.

## Introduction

Mitochondria are vital bioenergetic organelles responsible for essential energetic and metabolic processes in eukaryotes [1, 2]. Due to their evolutionary history, mitochondria have retained small genomes [3–5] that encode genes central to their functionality [6, 7]. MtDNA in several taxa, including many animals, is subject to a high mutation rate relative to the nucleus, and mutations in mtDNA cause cellular dysfunction, and are involved in a range of human diseases [8, 9]. As mtDNA is predominantly uniparentally transmitted [10], the question arises of how eukaryotes avoid the gradual accumulation of mtDNA mutations, known as Muller's ratchet [11].

The proportion of mutant mtDNA in a cell is usually referred to as heteroplasmy, and diseases are often manifest when heteroplasmy exceeds a certain level [12]. Eukaryotes may deploy a combination of strategies to generate cell-to-cell variability in inherited heteroplasmy [13–16], thus potentially generating offspring with heteroplasmies below a pathogenic threshold [9, 17, 18]. For instance, in mammalian development, a developing female produces a set of oocytes for the next generation. Through an effective 'genetic bottleneck', oocytes with different heteroplasmies are generated [15]. This range of heteroplasmies means that some cells may inherit a lower heteroplasmy than the mother's average. Across species, in concert with selection [19–24], this generation of variation allows shifts in heteroplasmy between generations [25, 26].

The mtDNA bottleneck has been suggested to incorporate a number of different mechanisms [27]. These include mtDNA depletion [28–30], and subpopulation replication [29, 31] in mammals, with a potential role for gene conversion in several other taxa [16], all coupled with random effects from partitioning of mtDNA at cell divisions. This partitioning is a focus of this report. Generally, when a cell divides, its mtDNA population will be partitioned between its daughter cells. Any deviation from precise deterministic partitioning (exactly half the mtDNA molecules of every genetic type go into each daughter) will likely lead to the daughter cells inheriting different heteroplasmy levels. The role of cell divisions—which may be asymmetric—in generating variability in important cell biological quantities has been studied with a growing body of recent theory, including effects on mRNA and protein levels [32, 33], cell size distributions [34], and general cell contents [35, 36] including organelles [16, 37, 38].

Models for mtDNA segregation have typically regarded the cellular mtDNA population as well-mixed [36], but the mitochondria containing the population have varied morphologies and dynamics in different cell types, with important consequent and emergent properties [39–42]. Mitochondria can form cell-wide networks in some cell types, undergoing fission and fusion [43–47]. These dynamic structures, together with mtDNA turnover, are linked to quality control and genetic dynamics of mtDNA populations [24, 48, 49]. In both somatic tissues [50–53], and the germline [16, 23], the balance between fusion, fission and selective degradation of individual dysfunctional mitochondria has emerged as an important influence on mtDNA populations [50, 52, 54–56].

The important effects of heterogeneous spatial distributions on noise in other cell biological contexts are being increasingly recognised [57]. The central importance of mitochondria in cell metabolism and bioenergetics mean that variability in their physical and genetic

inheritance can influence many other, also noisy, downstream processes [37, 58]. Quantitative progress modelling the spatial influence of these mitochondrial dynamics on mtDNA quality control is advancing [50–53, 56]. In particular, the role of network structure in generating cell-to-cell mtDNA variability via modulating mtDNA turnover has been addressed with recent stochastic modelling [16, 59]. These studies report that cell-to-cell mtDNA variability increases with the proportion of fragmented mitochondria in the cell, the rate of turnover, and the length of the cell cycle. Moreover, turnover itself may increase due to a highly fragmented network, with a highly fused network effectively masking mitochondria marked for degradation. However, the influence of mitochondrial network structure on the inheritance of mtDNA during cell divisions remains less studied. Although the mitochondrial network may fragment prior to cell division, allowing individual mitochondria to diffuse or actively mix [60], the network structure prior to division may exert substantial influence on the distribution of mtDNAs in the parent cell [43], ultimately reflected in daughter cell statistics. Partitioning at cell divisions even in the absence of spatial substructure can constitute an important source of cell-to-cell variability [35, 36]. In yeast, mtDNA inheritance occurs with finer-than-random (binomial) control over the number of mitochondrial nucleoids [61], suggesting that physical mechanisms must exist to exert this control. As mtDNA resides in nucleoids that are physically distributed—potentially heterogeneously—throughout the mitochondria of the cell [62], we set out to explore how different physical structures of mitochondria may shape mtDNA inheritance.

## Materials and methods

### Network simulation

We constructed a random network, modelling the arrangement of mitochondria inside a cell, via an elongation and branching process; network segments elongated deterministically with rate $e = 0.01$ and branched according to Poissonian dynamics with a given rate $k = 0.02$, and terminated if they hit the cell boundary. Network growth was initiated at a number of evenly-spaced seed points around the cell circumference (the first of which was randomly positioned in each simulation). Initial segments then grew perpendicular to the perimeter of a circular 2D cell, represented by the unit disc. Network growth proceeded until a predetermined network mass had been created; if all segments terminated before this mass was reached, we re-seeded the perimeter, and continued the growth process. We chose this threshold size as $U^* = 50r$ in units of cell radius $r$. Assuming a mitochondrial network tubule width of $\sim 0.4 \mu m$ and a cell radius of $40 \, \mu m$ (from an example mouse embryonic fibroblast image taken from [46]), this gives a mitochondrial area of $800 \mu m^2$ and a cell area of $\sim 5000 \mu m^2$, for a mitochondrial fraction of $\sim 16\%$. Although the mitochondrial content of different cell types varies substantially, this proportion is consistent with, for example, classical observations between $10 - 20\%$ in liver cells [63]. By changing the number of seed points from which segments grow, we tune the uniformity of the network structure: a high number of seed points yielded homogeneous network structures, whereas a low number of seed points often lead to heterogeneous network structures (see Fig 1). This growth process is not intended as a model of the true biophysics, where mitochondrial networks do not grow from nothing—we simply require a range of possible networks as the final result. The parameters were manually chosen to support a range of different network structures as seeding varied, and are not directly connected to real biophysical processes.

After simulating a network structure, we distributed mtDNAs in the cell. We considered two mtDNA types, wildtype $W$ and mutant type $M$, each a predetermined proportion of the mtDNA population, specified by $h$. Proportions $p$ of wild type mtDNA and $q$ of mutant type mtDNA molecules were then placed within the network according to a particular placement

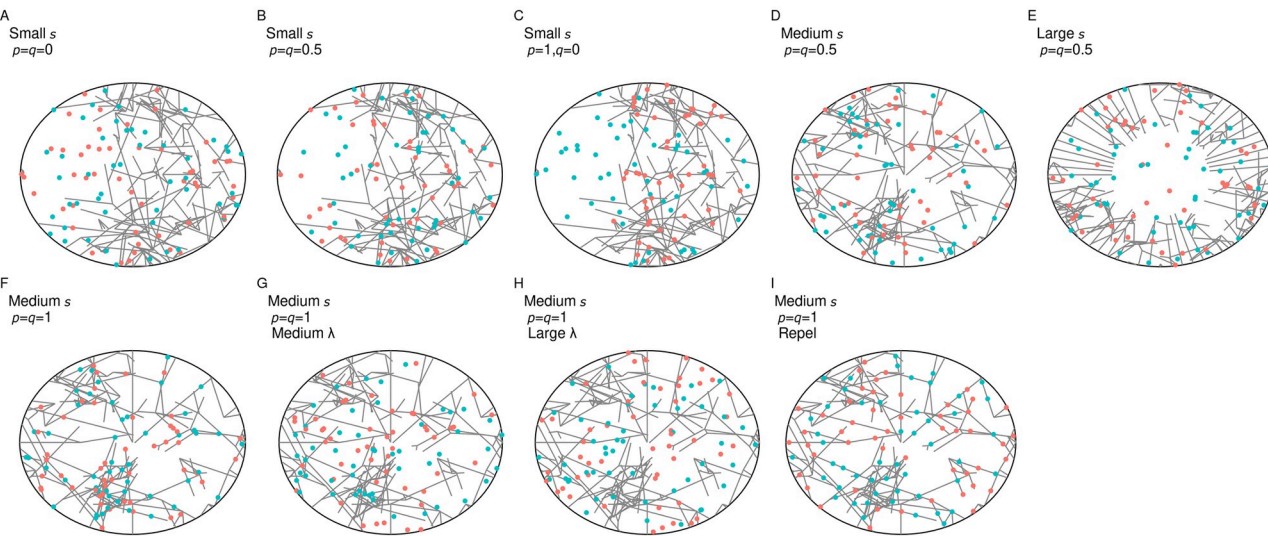

**Fig 1. Snapshots of computationally generated networks and mtDNA arrangements with tunable physical and genetic parameters.** Networks were generated by an elongation and branching process initialized from a number of seed points (*s*) uniformly distributed on the perimeter of the cell. Small seed numbers *s* (A-C) usually resulted in heterogeneous network structures, with marked differences in network density across the cell; for increasing seed numbers (D-F), networks were more uniformly distributed throughout the cell. Wild type (WT; red) and mutant type (MT; blue) mtDNA molecules were distributed into networks according to the proportions *p* (WT) and *q* (MT). In different model variants, diffusion with scale parameter λ was applied to mtDNAs prior to division to model network fragmentation and subsequent motion (G-H; original network shown for reference), and mtDNAs were placed with a repulsive interaction inducing greater-than-random spacing (I).

rule (see below). The remaining mtDNA molecules were randomly and uniformly distributed in the cell, modelling presence of fragmented organelles in the cytoplasm. First, we considered random and uniform placement of mtDNA within the network, in which every point of the network was equally likely to host an mtDNA molecule. Later we introduce a minimal inter-mtDNA distance within the network, hence enforcing spacing between mtDNA molecules through a mutual repulsion. This spacing is parameterised by *l*, the characteristic proportion of cell radius between mtDNA molecules. *l* = 0.05 for a 40 *μm* cell radius corresponds to a spacing of 2 *μm*, within the range of inter-nucleoid distances identified in Ref. [64].

Finally, we partition the cell and record the number of wildtype *W* and mutant *M* mtDNAs in one daughter, the heteroplasmy *h* = *m*/(*w* + *m*), as well as the proportion of network mass *u*. The process of cell division was modelled by recording only the network mass and mtDNA content of a fixed circular segment spanning an angle $\phi$ (with $\phi$ = 180˚ corresponding to symmetric cell division), randomly oriented with respect to the network seed points. We are particularly interested in the heteroplasmy variance $V(h)$ and copy number variance $V(N)$, where $N = W + M$, across many realisations of this system.

## Statistical models of mtDNA copy number and heteroplasmy

We consider $h = \frac{M}{W+M}$ and $N = W + M$ as our key variables. We assume that the parent cell's heteroplasmy level is $h \in [0, 1]$, with a total of $N_0$ mtDNA molecules. Thus there are $hN_0$ mutant molecules, and $(1 - h)N_0$ wild type molecules of mtDNA.

We ignore correlations between daughter cells and focus on a single daughter from a cell division. In the daughter, the copy number variance is

$$V(N) = V(W) + V(M) + 2\text{Cov}(W, M) \qquad (1)$$

Here $V(W)$ and $V(M)$ are the variances of wild-type and mutant mtDNA, respectively, and $\mathrm{Cov}(W, M)$ is the covariance of $W$ with $M$. The heteroplasmy variance, $V(h)$ does not follow a simple form as it deals with a ratio of random variables. Instead, we use either explicit sums for the moments as in the text:

$$
\begin{aligned}
E(f(h)) &= \sum_{W_c=0}^{w_c} P(W_c) \sum_{M_c=0}^{m_c} P(M_c) \int_0^1 P(U) dU \sum_{W_n=0}^{w_n} P(W_n|U) \\
&\times \sum_{M_n=0}^{m_n} P(M_n|U) f\left(\frac{M_n + M_c}{W_n + W_c + M_n + M_c}\right).
\end{aligned}
\tag{2}
$$

with mean $E(h)$ and variance $V(h) = E(h^2) - E(h)^2$, or a first-order Taylor expansion, finding that

$$
V_1(h) = h_M'^2 V(M) + h_W'^2 V(W) + 2h_M' h_W' \mathrm{Cov}(W, M)
\tag{3}
$$

The prefactors $h_M'$ and $h_W'$ are derivatives of the heteroplasmy level considered as a function of $W$ and $M$, and are model dependent (described below). Dividing $V_1(h)$ by $h(1 - h)$, we get the normalized heteroplasmy variance, $V_1'(h)$, which conveniently removes some of the dependence on $h$. We also considered higher order terms in this expansion (see S1 Appendix).

**The null hypothesis: No mtDNA placement in network.**   As our null hypothesis, we considered a binomial segregation model for mtDNA [15]. In this model, no network structure exists and no active mechanisms contribute to the distributions of mtDNA (of either type) to the daughter cell. Supposing that the cell divides such that the daughter consists of a proportion $p_c$ of the parent cell volume, we supposed

$$
M \sim \mathrm{Bin}(hN_0, p_c) \ \text{ and } \ W \sim \mathrm{Bin}((1 - h)N_0, p_c)
\tag{4}
$$

The copy number variance of the daughter is then, from the binomial distribution,

$$
V(N) = p_c(1 - p_c)N_0
\tag{5}
$$

To find the heteroplasmy level variance, we calculated the variances of $W$ and $M$, and their corresponding derivatives; the covariance of $W$ and $M$ is zero in this case. A detailed derivation of $V_1(h)$ is presented in S1 Appendix, the result of which is

$$
V'(h) = \frac{1 - p_c}{p_c} \frac{1}{N_0}
\tag{6}
$$

This is our null case, with nothing actively influencing the placement of mtDNAs within the cell. If this is the case, apportioning of mtDNA to daughter cells is binomial. The familiar expression of $1/N_0$ is recapitulated for symmetric cell divisions, i.e., with $p_c = 1/2$, in which case $V'(h) = 1/N_0$.

**Random mtDNA placement in network.**   Following intuition and preliminary observation of our simulations, we model the proportion $u$ of network mass inherited by the smaller daughter as beta distributed variable $U$, with mean $E(U)$ and variance $V(U)$. Expected network inheritance $E(U)$ will simply be $p_c$, the proportion of cell volume inherited; $V(U)$ will depend on the spread of the network through the cell, and constitutes a fit parameter in comparing this statistical model to simulation. Hence a particular value $u$ is drawn from the beta distribution, $\mathrm{Beta}(\alpha, \beta)$ with mean $E(U) = \alpha/(\alpha + \beta) = p_c$ and variance $V(U) = \frac{\alpha\beta}{(\alpha+\beta)^2(\alpha+\beta+1)}$.

We now write $W_n$, $W_c$ respectively for the number of wildtype mtDNAs placed in the network and randomly spread in the cytoplasm, and $M_n$, $M_c$ likewise for mutant mtDNA. $W_c$ and $M_c$ are assumed to follow the binomial partitioning dynamics above. Assuming that mtDNAs

in the network are randomly positioned therein, we draw a $u \sim \text{Beta}(\alpha, \beta)$ to reflect the network proportion inherited by the smaller daughter, and write

$$W_n \sim \text{Bin}(w_n, u)$$

$$W_c \sim \text{Bin}(w_c, p_c)$$

$$M_n \sim \text{Bin}(m_n, u)$$

$$M_c \sim \text{Bin}(m_c, p_c)$$

$$(7)$$

where $w_n = p(1 - h)N_0$, $w_c = (1 - p)(1 - h)N_0$, $m_n = qhN_0$, $m_c = (1 - q)hN_0$.

The mean and variance of $N$ are readily derived using the laws of iterated expectation and total variance to account for the compound distribution of mtDNA in the network (S1 Appendix). To estimate heteroplasmy variance, we combine the (co)variances of the different mtDNA types as described in S1 Appendix).

**Repulsive mtDNA placement in network.**   Next, we considered the case where mtDNAs placed in the network are not randomly positioned, but instead experience a repulsive interaction, and thus adopt a more even spacing. Capturing this picture perfectly with a statistical model is challenging; instead, we use the following picture. The proportion of inherited network mass $u$ consists of a finite number of 'spaces', each of which can be occupied by at most one mtDNA molecule. Choose a number of spaces to fill, then sample mtDNA molecules from the available pool without replacement, assigning each drawn mtDNA to the next unoccupied network space. In this case, the final population of mtDNAs in the network is described by the hypergeometric distribution. We draw $u \sim \text{Beta}(\alpha, \beta)$ to reflect the network proportion inherited by the smaller daughter, and write

$$W_n \sim \text{Hypergeometric}(w_n + m_n, w_n, \lfloor u/l \rfloor)$$

$$W_c \sim \text{Bin}(w_c, p_c)$$

$$M_n \sim \lfloor u/l \rfloor - W_n$$

$$M_c \sim \text{Bin}(m_c, p_c)$$

$$(8)$$

We can once again use the laws of total variance and iterated expectation (see S1 Appendix) to estimate heteroplasmy and copy number behaviour. As previously discussed, this model has several shortcomings and is only expected to match qualitative behaviour (see S1 Appendix).

## Results

### Network inclusion with genetic bias increases cell-to-cell variability

To build intuition about the influence of network structure on mtDNA inheritance, we first considered a simple computational model for the spatial distribution of mitochondria and mtDNA within a cell (see Methods). This model consists of a random network structure, with a tunably heterogeneous distribution, simulated within a circular cell (see Fig 1). The model resembles a general interpolation between the 'clustered' and 'spaced' models for mitochondrial and nucleoid distributions in Ref. [61], with a parameter allowing different structures between these limiting cases to be generated. This 2D model will capture several common cell morphologies where one dimension is much smaller than others; furthermore, the aspects of the model that would change in higher dimensions are those controlled by the parameters of our model and so can readily be tuned to any required circumstances. The mother cell has a

fixed population of $N_0$ mtDNAs, a proportion $h$ of which are mutants ($h$ is heteroplasmy). The model structure is not intended to perfectly match the details of real mitochondrial networks, but rather as a general framework to understand spatial substructure in the cell. To compare roughly with mammalian cells, assuming a mitochondrial tubule width of $1\mu m$ corresponds to our model networks occupying roughly 1/6 of the cell volume, consistent with, for example, mammalian liver cells [63] (see Methods).

To reflect the fact that different mtDNA types may have different propensities to be included in the reticulated network [54], we use $p$ and $q$ to describe the proportions of wildtype and mutant mtDNAs respectively that are contained in the network; the remainder are in fragments in the cytoplasm (Fig 1). Hence, $p > q$ means that wildtype mtDNAs are more likely to be contained in the network and mutant mtDNAs are less likely to be included; $p = q$ means that both types are equally likely to be in a networked state. Network placement can follow various rules (described below) and mtDNA positions may be subsequently perturbed, but we begin with random placement in the network and no subsequent motion prior to division. We then divide the model cell and explore the statistics of mtDNA copy number and heteroplasmy in the daughter cells. We first consider symmetric partitioning, so that the initial cell is physically halved to produce two daughters; we relax this assumption and consider asymmetric partitioning later.

To explore the influence of mtDNA network placement in the parent cell on the mtDNA statistics of the daughter, we varied network inclusion probabilities $p$ and $q$ (Fig 1A–1C) and network heterogeneities (Fig 1C–1E) and observed the variances of copy number $V(N)$ and heteroplasmy $V(h)$ in daughter cells after division (Fig 2). These results are for an illustrative case where $N_0 = 100$, and the behaviour is qualitatively preserved for larger mtDNA populations (S1 Fig shows the equivalent results for $N = 1000$ mtDNAs). We found a clear pattern that $V(h)$ takes minimum values when $p = q$, that is when network inclusion probabilities are equal for mutant and wildtype mtDNA. When the two differ, $V(h)$ increases, with highest values occurring when the majority mtDNA type is exclusively contained in the network and the minority type exclusively in the cytoplasm.

This result may seem counterintuitive at first glance: when the network is highly heterogeneous, it might be expected that including all mtDNAs there would maximise variance. This is true for copy number variance (Fig 2), but not for heteroplasmy variance, because including both types equally induces correlation in their inheritance and lowers variance. The maximum $V(h)$ is achieved by embedding the majority type in the high-variance network and having the minority type in the uncorrelated cytoplasm.

## Statistical models of mtDNA inheritance capture the variance induced by cell divisions

To further explore this behaviour, we constructed a statistical model for this inheritance process (see Methods). The system begins with a circular mother cell, with deterministic mtDNA contents described as above by the total copy number $N_0$, the proportion of mutants (heteroplasmy) $h$, and $p$, $q$ respectively the proportion of wildtype and mutant mtDNAs contained in a network (as opposed to fragmented organelles). The network is constructed by random processes, and it and the fragmented organelles will both in general be unevenly distributed through the mother cell. The cell then divides according to a given division rule, corresponding to a sector (half the circle, in the symmetric case) being removed as one daughter and the remaining sector forming the other. Then we consider four state variables describing the mtDNA population of a daughter cell after the mother divides: $W_n$, $W_c$, $M_n$, $M_c$ for the wildtype ($w$) and mutant ($m$) mtDNAs contained in a reticulated mitochondrial network ($_n$) or in fragmented mitochondrial elements in the cytoplasm ($_c$). An additional variable $U$ describes

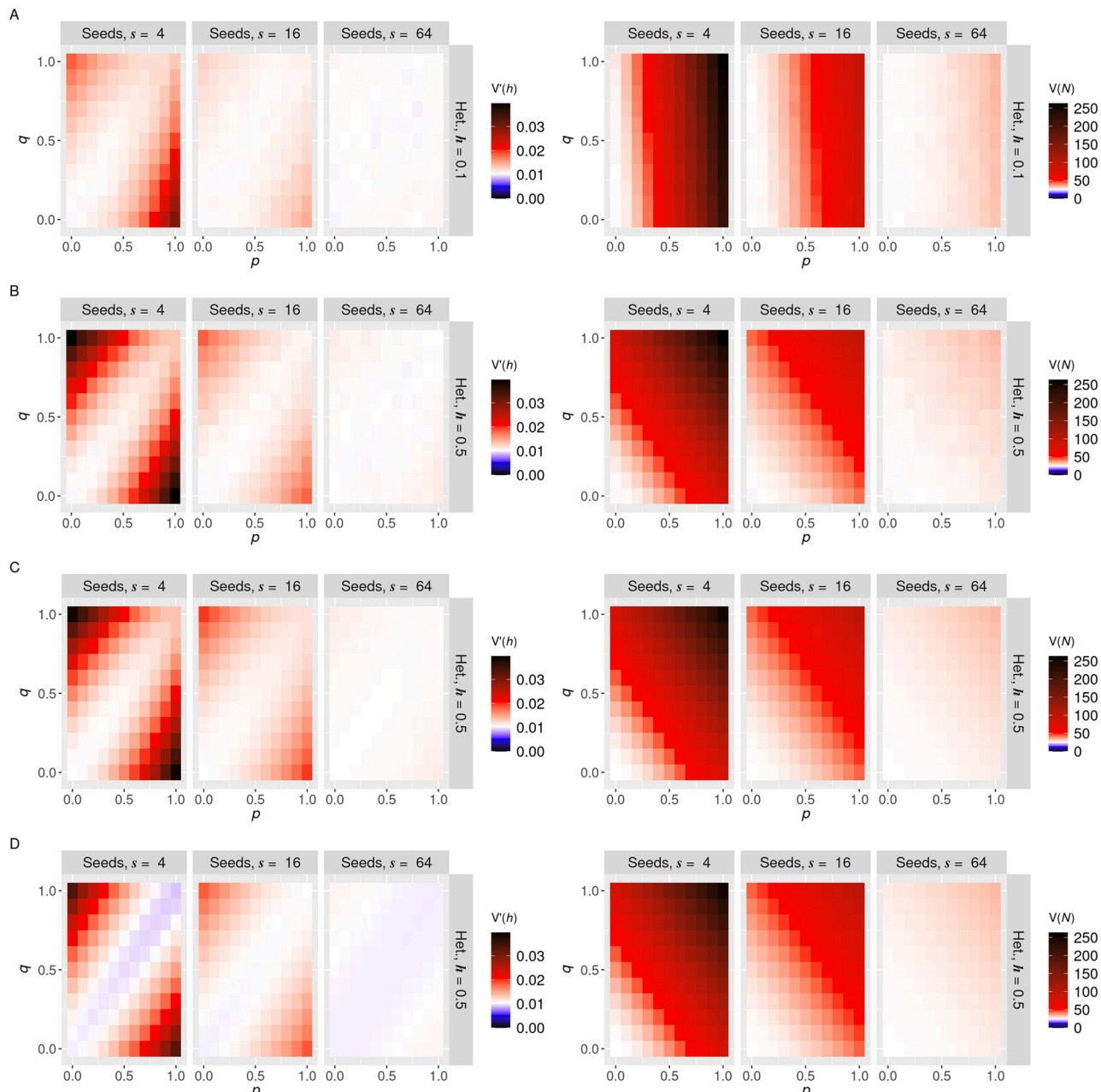

**Fig 2. Genetic bias and physical heterogeneity in mitochondrial networks induce cell-to-cell mtDNA variability in symmetric cell divisions.**
Normalised heteroplasmy variance $V'(h)$ (left column) and copy number variance $V(N)$ (right column) for $N = 100$ mtDNA molecules randomly distributed in networks. **A**: simulations for $h = 0.1$; **B**: simulations for $h = 0.5$; **C**: sum over state variables for $h = 0.5$; **D**: first-order Taylor expansion for $h = 0.5$. The three columns for each panel give decreasing network heterogeneity, expressed via different seed numbers, 4, 16 and 64 (more seed points give a more homogeneous network). In each panel, variances are given for different values of wild and mutant type network inclusion parameters $p$ (horizontal axis) and $q$ (vertical axis). White baseline reflects the null case from the analytic sum without any network inclusion.

the proportion of the mother's network mass inherited by the daughter cell. Given a particular value $u$ for this proportion, the mtDNA profile inherited by the daughter follows

$$W_n \sim \mathrm{Bin}(w_n, u)$$

$$W_c \sim \mathrm{Bin}(w_c, 1/2)$$

$$M_n \sim \mathrm{Bin}(m_n, u)$$

$$M_c \sim \mathrm{Bin}(m_c, 1/2)$$

(9)

where $w_n = p(1 - h)N_0$, $w_c = (1 - p)(1 - h)N_0$, $m_n = qhN_0$, $m_c = (1 - q)hN_0$. We model the proportion $u$ of network mass inherited by a daughter with a beta-distributed variable $U$, with variance $V(U)$ allowed to vary to describe different partitioning regimes of network mass. To compare to simulations, we fit the beta distribution parameters to match the simulated inherited network variance. As $W_n$ and $M_n$ are then drawn from the compound distribution that is binomial with a beta-distributed probability, they follow beta-binomial distributions. We are interested in the inherited copy number $N = W_n + W_c + M_n + M_c$ and heteroplasmy $h = (M_n + M_c)/N$.

We can numerically extract moments for the system through summing over state variables and calculating expectations, for example,

$$
\begin{aligned}
E(f(h)) &= \sum_{W_c=0}^{w_c} P(W_c) \sum_{M_c=0}^{m_c} P(M_c) \int_0^1 P(U) dU \\
&\times \sum_{W_n=0}^{w_n} P(W_n|U) \sum_{M_n=0}^{m_n} P(M_n|U) f\left(\frac{M_n + M_c}{W_n + W_c + M_n + M_c}\right).
\end{aligned}
$$

(10)

Figs 2 and 3 demonstrates good correspondence between simulations and statistics using Eq 10 for copy number and heteroplasmy variance ($V(X) = E(X^2) - E(X)^2$). However, as these large sums do not admit much intuitive analysis, we sought other approaches to learn the forms of pertinent statistics of the inherited mtDNA population.

The mean and variance of inherited number $N$ are readily derived using the laws of iterated expectation and total variance to account for the compound distribution of networked mtDNA (S1 Appendix):

$$V(N) = \frac{N_0}{4} + \kappa N_0(\kappa N_0 - 1)V(U)$$

(11)

where $\kappa = p(1 - h) + qh$ denotes the proportion of total mtDNA placed in the network. Hence $V(N)$ experiences an extra, $V(U)$-dependent term in addition to the expected $N_0/4$ result for a purely binomial distribution. This term is quadratic in the proportion $\kappa$ of mtDNA in the network. This expression well captures the results from simulation and Eq 10 (Fig 2).

For heteroplasmy variance, as the ratio of random variables, we cannot extract an exact solution and must instead use a Taylor-expanded approximation (see Methods) to obtain

$$V'(h) \simeq V'_1(h) = \frac{1}{N_0} + 4V(U)\big(h(1 - h)(p - q)^2 - (ph + q(1 - h))/N_0\big)$$

(12)

Eq 12 allows some informative analysis. First, we qualitatively see that an additional, $V(U)$-dependent term is introduced compared to the binomial case (which gives $1/N_0$), illustrating the influence of network heterogeneity on heteroplasmy variance. For large $N_0$, this network term is dominated by the first term in brackets in Eq 12, which is quadratic in $(p - q)$, the

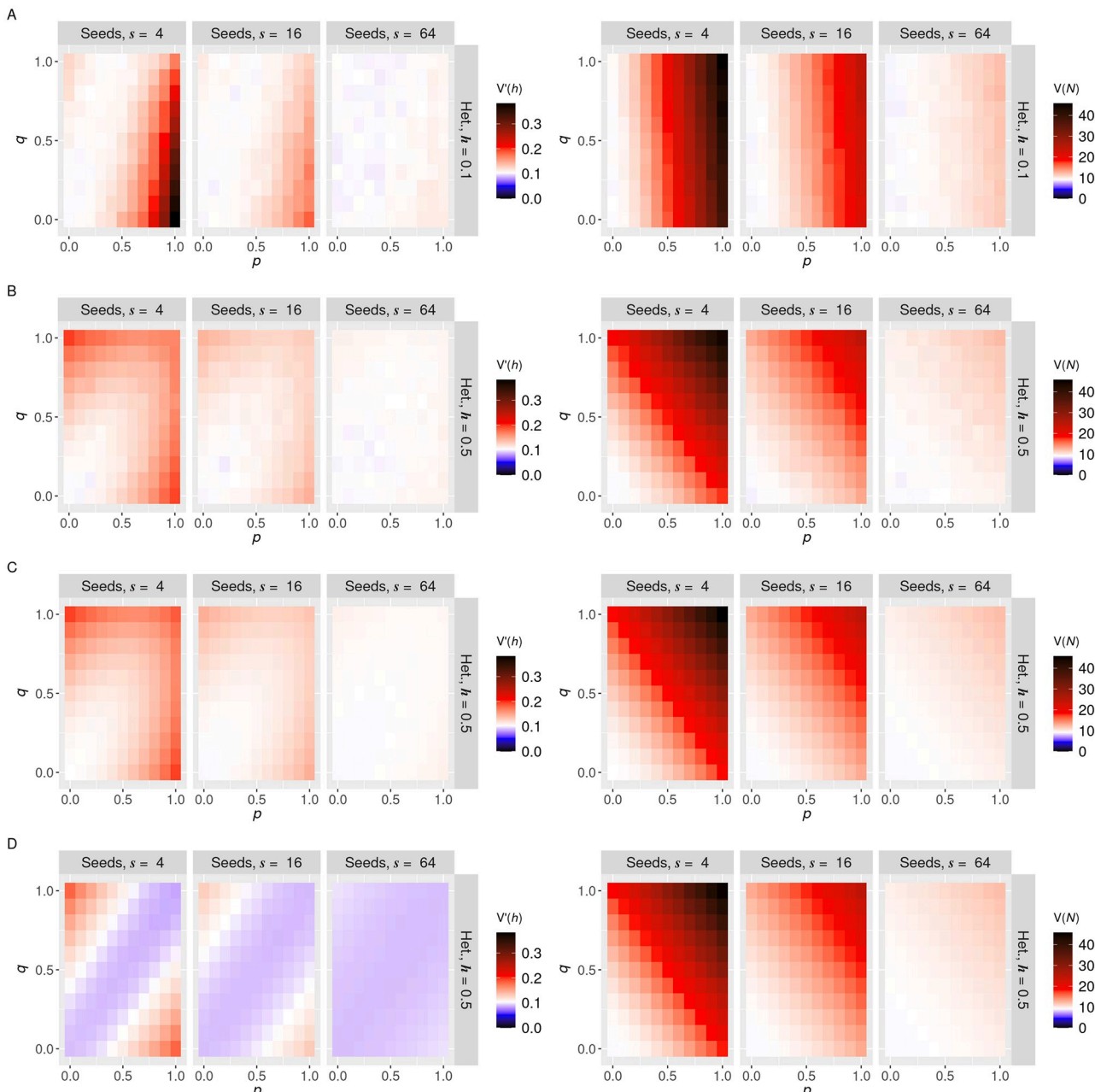

**Fig 3. Asymmetric cell divisions can induce large cell-to-cell variability in mtDNA quality.** Normalised heteroplasmy variance $V'(h)$ (left column) and copy number variance $V(N)$ (right column) for mtDNA randomly distributed in networks. The daughter of interest inherits 10% of the parent cytoplasm. **A**: simulations for $h = 0.1$; **B**: simulations for $h = 0.5$; **C**: sum over state variables; **D**: first-order Taylor expansion. The three columns for each panel give decreasing network heterogeneity, expressed via different seed numbers, 4, 16 and 64 (more seed points give a more homogeneous network). In each panel, variances are given for different values of wild and mutant type network inclusion parameters $p$ (horizontal axis) and $q$ (vertical axis). White baseline reflects the null case from the analytic sum without any network inclusion.

difference in inclusion probabilities for the different types of mtDNA. For $p \neq q$, the network is genetically biased towards one of the types, to which there is associated an increase in $V'(h)$. For $p = q$, the network is unbiased, with associated prediction $V'_1(h) = \frac{1}{N_0} - \frac{4pV(U)}{N_0}$—that is, a small negative change from the binomial case. However, this negative shift is in fact an artefact

of the imperfect Taylor approximation we use (see below and S1 Appendix), and the $p = q$ case in fact resembles the binomial case with a slight increase (captured by a higher-order approximation, see S1 Appendix) at higher $p$ (Fig 2).

The more useful prediction under this approximate model concerns the maximum normalised heteroplasmy variance achievable—when the majority mtDNA type is completely contained in the network and the minority type completely excluded from it—is given for example by setting $p = 1$, $q = 0$ for $h \leq 0.5$:

$$V_1'(h) = \frac{1}{N_0} + 4V(U)(h(1 - h) - h/N_0), \tag{13}$$

with a $V(U)$-dependent term scaled by $h(1 - h)$ (in most cases the $h/N_0$ term will be negligible), illustrating that genetic bias in network inclusion can substantially increase heteroplasmy variance in proportion to inherited network variability.

This picture does not completely capture the simulation and exact results, where we see a small increase in (normalized) heteroplasmy variance for non-biased increases in inclusion probabilities, instead of a decrease. This reflects the approximate nature of the Taylor expansion process used to derive Eq 12; in S1 Appendix we show that a second-order expansion provides a compensatory diagonal term (S4 Fig), but in general we will need further terms in the expansion to perfectly match the true behaviour. In S1 Appendix we further show that all higher-order moments and covariances of the mtDNA copy numbers are well captured by theory; it is their combination into an estimate for the moments of a ratio (heteroplasmy) that leads to departures from analytic and simulated results here. We will observe below that this Taylor expansion approach (which has been successfully employed previously [27, 38, 65]) also fails to provide accurate estimates for more instances of this system—generally, accounting for physical structure induces correlations between mtDNA types that are hard to capture with the Taylor expression (see Discussion).

## Asymmetric cell divisions induce more mtDNA variability

The above model has assumed that the cell divides symmetrically, with half the cell volume inherited by each daughter. To generalise to asymmetric cell divisions, we next asked how the proportion of inherited cell volume $p_c$ ($p_c = 1/2$ in the symmetric case) influences the mtDNA statistics in daughters. Clearly, the expected copy number will differ if the inherited proportion differs. The expressions above generalise to

$$V(N) \quad = \quad N_0 p_c(1 - p_c) + \kappa N_0(\kappa N_0 - 1)V(U) \tag{14}$$

$$V_1'(h) \quad = \quad \frac{1 - p_c}{p_c}\frac{1}{N_0} + \frac{V(U)}{p_c^2}\left(h(1 - h)(p - q)^2 - (ph + q(1 - h))/N_0\right), \tag{15}$$

with the result that asymmetric cell divisions can generate large increases in cell-to-cell variability for the smaller daughters (Fig 3). Small number effects are at play here, with a smaller sampling of the initial cell inevitably leading to greater relative variance. A decrease in $p_c$ from the symmetric case $p_c = 1/2$ to $p_c = 0.1$ results in a near order-of-magnitude increase in the maximum normalised heteroplasmy variance. Asymmetric divisions also further challenge the Taylor expansion approach, with a systematic underestimation of heteroplasmy variance apparent using this approximation (copy number variance remains well captured).

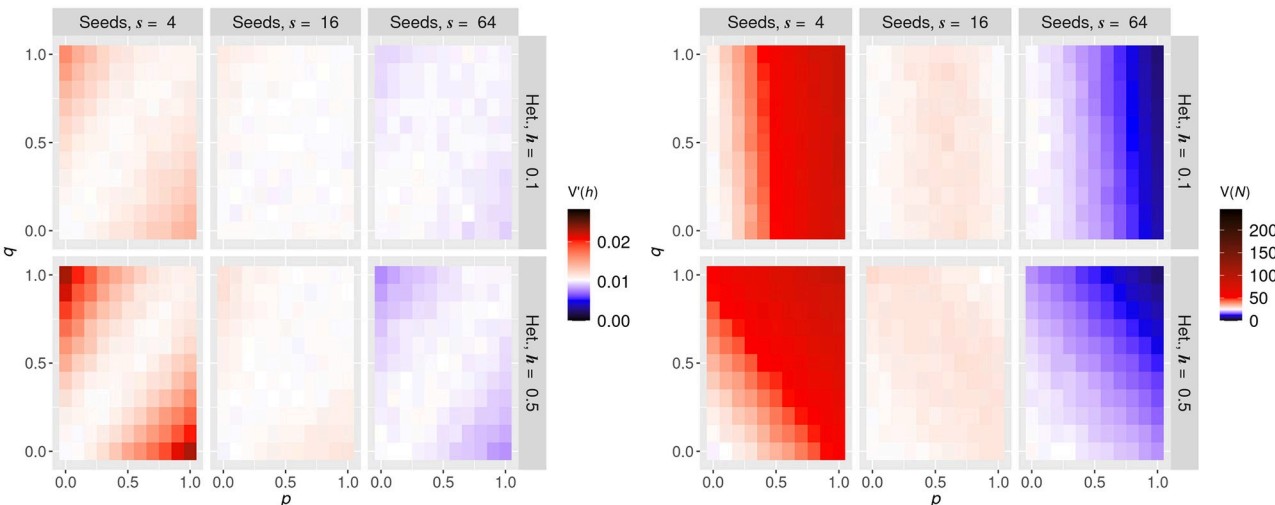

**Fig 4. Active spread of mtDNAs in the network can increase and decrease cell-to-cell variability from cell divisions.** Simulated normalised heteroplasmy variance $V'(h)$ (left column) and copy number variance $V(N)$ (right column) for mtDNAs with mutual repulsion with radius $l = 0.1$ within the network, under symmetric cell divisions. Rows show different values of initial mutant proportion, with $h = 0.1$ in the top row and $h = 0.5$ in the bottom row. The three columns for each panel give decreasing network heterogeneity, expressed via different seed numbers, 4, 16 and 64 (more seed points give a more homogeneous network). In each panel, variances are given for different values of wild and mutant type network inclusion parameters $p$ (horizontal axis) and $q$ (vertical axis). White baseline reflects the null case from the analytic sum without any network inclusion.

## MtDNA self-avoidance tightens mtDNA copy number control and can reduce heteroplasmy variance

We next asked whether the variances of copy number $V(N)$ and heteroplasmy $V(h)$ could be reduced below their 'null' binomial value through cellular control. To this end, we modelled self-avoidance of mtDNA molecules within the network (Fig 1I), reasoning that such controlled arrangement may allow a more even spread of mtDNAs within the network, and correspondingly lower variability. To accomplish this self-avoidance within our model, we enforce a 'halo' of exclusion around each mtDNA placed within the network, so that another networked mtDNA cannot be placed within a distance $l$ of an existing one. $l$ is measured in units of cell radius, so, for example, $l = 0.05$ enforces that mtDNAs must be separated by a minimum of 5% of the cell radius. The results are shown in Fig 4 for $l = 0.1$ (pronounced inter-mtDNA spacing to demonstrate the effects) and S3 Fig for $l = 0.05$ (reflecting a wider range of biological examples, see Methods).

Copy number variance $V(N)$ is decreased substantially by self-avoidance (Fig 4). In the case of a homogenous network and high proportions of mtDNA network inclusion, this decrease can readily extend below the binomial null model, allowing more faithful than binomial inheritance, as reported in yeast [61]. This sub-binomial inheritance requires both an even network distribution and mtDNA self-avoidance, and hence two levels of active cellular control —following the findings in Ref. [61]. Under these circumstances, mtDNA molecules are evenly spread throughout the cell volume, and their inheritance approaches a deterministic proportion of the inherited volume fraction $p_c$.

The effect of mtDNA self-avoidance on heteroplasmy variance is more complicated. For highly heterogeneous network distributions, heteroplasmy variance follows the same qualitative pattern as for the non-repulsive case, with higher variances achieved when network inclusion discriminates wildtype and mutant types. However, for homogeneous network distributions, the opposite case becomes true. Here, network inclusion discrimination *lowers*

the heteroplasmy variance induced by cell divisions. This possibly surprising result arises because of covariance effects. When both genotypes are equally included in the network, the variance of both copy numbers is relatively low, but there is substantial (negative) correlation because of mutual repulsion. Regions with many wildtypes will contain few mutants and vice versa, effectively inducing spatial clustering of the genetic populations within the homogeneous network, and hence increasing heteroplasmy variance upon partitioning. The first-order Taylor approximations in our model, though not a perfect representation, demonstrate that the $Cov(W, M)$ covariance term in $V(h)$ has a negative coefficient—so a negative covariance will increase heteroplasmy variance. When one genotype but not the other is in the network, this correlation effectively vanishes, and the heteroplasmy variance can become lower than binomial because, while one genotype is binomially distributed, the other is more evenly spread.

The heteroplasmy variance induced by cell divisions can be controlled to sub-binomial levels in the case of self-avoidance, strong discrimination, and a homogeneous network distribution. Notably, it is possible for the cell to control copy number variance below the binomial limit while also generating heteroplasmy variance, without biased network inclusion (for example, in the $n = 64$, $h = 0.5$ cases in Fig 4), reflecting a potentially beneficial case for implementing a genetic bottleneck without challenging overall mtDNA levels.

Analytic progress is more challenging for this case, but an imperfect statistical model (S2 Fig; see Methods) can begin to capture some of the qualitative behaviour. The model correctly predicts the direction of change of copy number variance for various network structures, and the capacity to control variance below the binomial value, but the magnitudes of predicted variances are more extreme than those observed in simulation. This discrepancy arises because, to retain tractability, the algebraic model imposes an even spread of mtDNAs more strictly than is applied in the simulation (where this imposition is limited for numerical reasons). The range of variance values in the simulation is thus more limited than those that emerge from model predictions, although the trends of behaviour with governing variables are largely consistent.

## Diffusion of mtDNA relaxes statistics towards their null value

The mitochondrial network fragments before cell division. This fragmentation gives a time window during which mtDNAs that were previously constrained by the network structure can diffuse away from their initial position. In the limit of infinite diffusion, network structure will be forgotten and the mtDNA population will be randomly and uniformly distributed throughout the cell, leading to binomial inheritance patterns. To connect with recent literature highlighting the importance of this pre-division motion [60], we next investigated how limited amounts of diffusion away from the initial structure influence the patterns of mtDNA inheritance. To this end, we fragment the network structure in our model so that each mtDNA molecule is free to diffuse (in an individual organelle). Diffusion is applied through 100 normally-distributed random steps of width λ (in units of cell radius), so that mtDNA molecules undergo random walks with expected total displacement around 10λ (though not exactly this value, due to boundary conditions). This model (Fig 1G and 1H) mirrors findings that mitochondria may undergo a series of directed bursts of motion from active randomization prior to cell division [60].

Fig 5 shows $V'(h)$ for random placement of mtDNA in the network (left column) and repulsive placement of mtDNA in the network (right column) for 4, 16, and 64 seed points (more seed points give a more homogeneous network), respectively, in rows from the top to the bottom. Our results indicate that directed bursts of motion of the mitochondria indeed work to

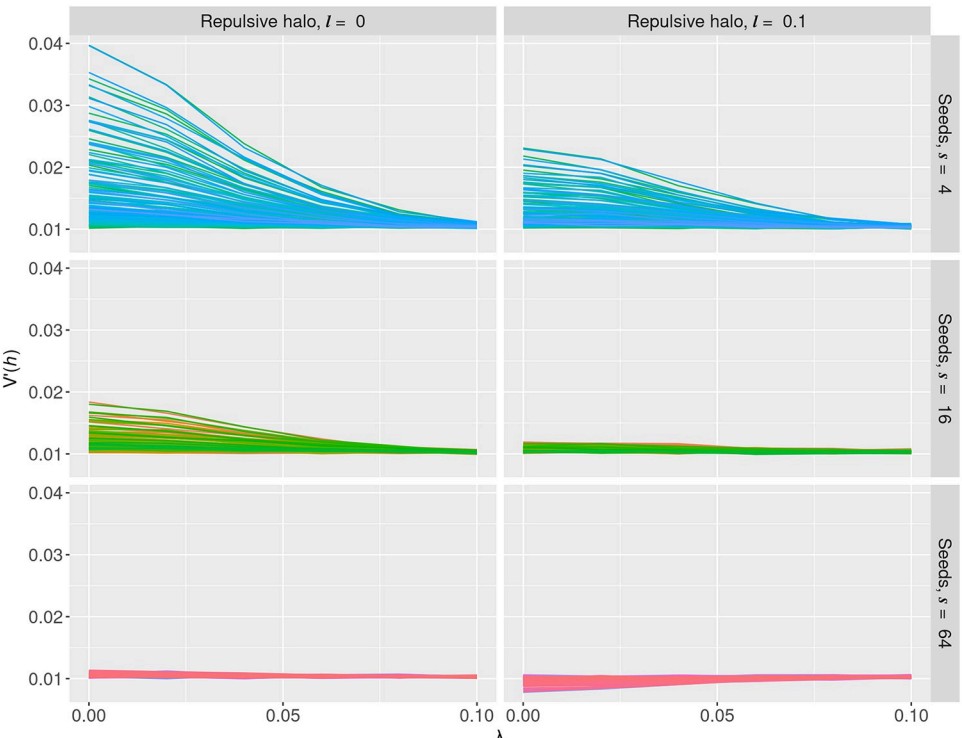

**Fig 5. Network fragmentation and diffusion reverse the variance induced by mitochondrial networks.** Normalised heteroplasmy variance $V'(h)$ after repeatedly perturbing each mtDNA molecule with mean displacement λ from their original positions, resulting in a total mean displacement of around 10λ. The left column shows $V'(h)$ for random spread of mtDNA in the network (i.e., with a repulsive radius of $l = 0$); the right column shows $V'(h)$ for active spread of mtDNA in the network with a radius of $l = 0.1$. The three rows show decreasing levels of network heterogeneity, expressed via different seed numbers (more seed points give a more homogeneous network). As the diffusion strength increases, effects on cell statistics due to the network is washed away regardless of the underlying mtDNA distribution model.

remove the effects of the network on daughter cell statistics, resulting in binomial segregation. Notably, there are cases in which extremely heterogeneous mitochondrial networks can leave an imprint on mtDNA inheritance despite high diffusion strength (top row).

## Discussion

We have demonstrated that a cell's mitochondrial network structure can control cell-division-induced variability in both mtDNA copy number and heteroplasmy inheritance, in both directions (Fig 6). When different mtDNA genotypes have different propensities for network inclusion, random arrangement in a heterogeneous network generates (much) more variability than could be achieved through random cytoplasmic arrangement alone. On the other hand, ordered arrangement in a homogeneous network can control heteroplasmy variance below the binomial level expected from random partitioning. In concert, homogeneous network structure dramatically reduces copy number variance to sub-binomial levels (as observed experimentally in Ref. [61]), and heterogeneous network structure correspondingly increases it. Notably, it is possible to tightly control copy number while generating heteroplasmy variance —a strategy that may be useful in implementing a beneficial genetic bottleneck to segregate mtDNA damage.

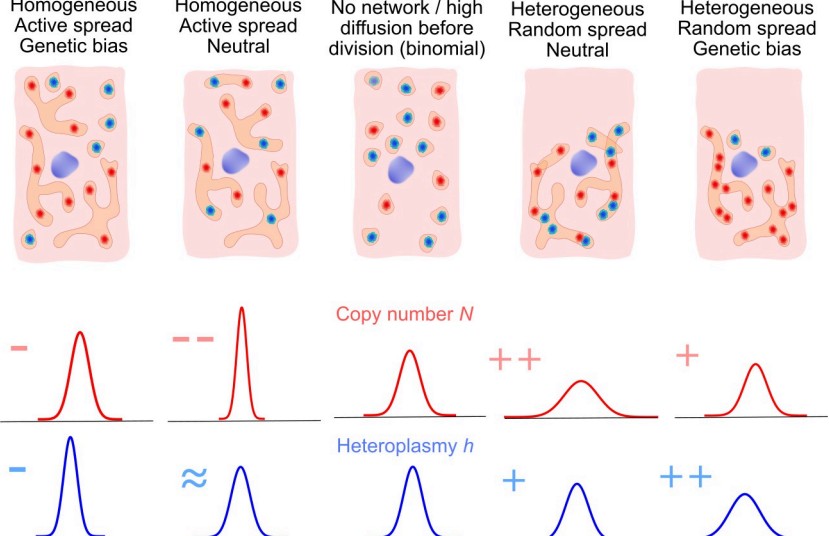

**Fig 6. Illustration of the range of influences of network structure and contents on mtDNA statistics from partitioning.** MtDNA molecules distributed randomly through the cytoplasm are segregated binomially (centre); the influence of network structure (heterogeneous or homogeneous distributions) and mtDNA inclusion (genetically neutral or biased) changes cell-to-cell variability in different ways: Depending upon genetic bias and network heterogeneity, networks can both increase and decrease cell-to-cell mtDNA variability in copy number and heteroplasmy. Genetic bias for network inclusion can modulate (physical) copy number statistics, by influencing the total number of mtDNA molecules that are evenly spread through the cell.

How do mitochondria in different taxa and tissues correspond to the different regimes in our model? In animals, mitochondrial structure is highly varied, from largely fragmented organelles (low $p$ and $q$ in our model) to highly reticulated networks (high $p$ and/or $q$), with mtDNA nucleoids moving freely throughout the mitochondrial reticulum, leading to a random distribution of mtDNA within the network [66, 67]. Animal mitochondria are expected to fragment prior to cell divisions, with a spectrum of active randomization mechanisms [60, 68], captured by the diffusion behaviour in our model. Fungal mitochondria, on the other hand, are often inherited without fragmenting the network, which remains in its reticulated state through cell division [43]. *Saccharomyces cerevisiae* populations have been shown to clear heteroplasmy within a few generations, with relatively heterogeneous networks and semi-regular spacing of mtDNA [24]. This situation corresponds to the 'repulsive' version of our model, inducing heteroplasmy variance (helping to clear heteroplasmy) while controlling copy number. In plants, mitochondria normally exist in a fragmented state (low $p$ and $q$) [40, 69]. An intriguing exception to this is the formation of a reticulated network prior to division in the shoot apical meristem—the tissue that gives rise to the aboveground germline [18]. This formation could be the signature of network structure being employed to shape mtDNA prior to inheritance—although this employment is also likely to involve facilitating recombination [16]. One useful extension to this model would be to consider spatial correlations in mtDNA type, so that wildtype and mutant mtDNAs are more likely to be close to others of the same genotype. This correlation would arise from clonal expansion of mtDNAs in a cell with limited subsequent motion, and would potentially further increase heteroplasmy variance generated through divisions. Another extension would be to consider an explicit model for network growth between cell divisions, which would allow a tighter connection to real network topologies and would support coupling between network state and cell cycle progression. Previous

work has modelled linked mitochondrial development and cell cycle progression [37] and more general concentration homeostasis over cell divisions [70]; coupling these approaches with mitochondrial spatial structure would allow a more universal theory.

One technical observation from our work is that the Taylor expansion method for approximating heteroplasmy variance, often employed in mtDNA models [27, 65], has several shortcomings in the face of network structure and other physical phenomenology that induce correlations between mtDNA types. Higher-order terms in the expansion do not immediately fix the discrepancies from simulation; we conclude that the series is likely slow to converge in these cases. Our model does successfully capture all the moments and covariances of the quantities of interest (see S1 Appendix)—so, for example, all pertinent statistics of the number of wildtype mtDNAs can reliably be extract. It is the combination of these statistics into an approximation for a ratio (heteroplasmy) that is unreliable. We advocate the use of analytic expressions (like our exhaustive sum over states) and explicit simulation to check the validity of such results in future contexts (in a sense paralleling the careful consideration of moment-based methods for stochastic chemical kinetics [71]).

Our observations on how network structure influences genetic population structure stand in parallel with the many other phenomena associated with the physical and genetic behaviour of mitochondria. Mitochondrial network structure and dynamics likely fulfil many purposes [45], including contributing to mtDNA quality control [54, 55] via facilitating selection. Here, we assume that selection occurs (if at all) between cell divisions, focussing rather on the behaviour at cell divisions. Previous modelling work has demonstrated the capacity of mitochondrial network structure to shape mtDNA genetics through ongoing processes through the cell cycle [16, 59]; other work has considered the behaviour of controlled mtDNA populations across divisions without considering how that control may be physically manifest [36, 38]. We hope that our models here help bridge the gap between these pictures of mitochondrial spatial dynamics between, and well-mixed behaviour at, cell divisions across a range of eukaryotic life.

## Supporting information

**S1 Appendix. Mathematical derivations.** Statistical modelling and moment calculations for physical and genetic distributions of inherited mitochondria.
(PDF)

**S1 Fig. Cell-to-cell mtDNA variability for larger mtDNA populations.** Following Fig 2 in the main text, but for $N_0 = 1000$ mtDNAs rather than $N_0 = 100$. Normalised heteroplasmy variance $V'(h)$ (left column) and copy number variance $V(N)$ (right column) for mtDNA randomly distributed in networks. **A**: simulations for $h = 0.1$; **B**: simulations for $h = 0.5$; **C**: sum over state variables for $h = 0.5$; **D**: first-order Taylor expansion for $h = 0.5$. The three columns for each panel give decreasing network heterogeneity, expressed via different seed numbers, 4, 16 and 64 (more seed points give a more homogeneous network). In each panel, variances are given for different values of wild and mutant type network inclusion parameters $p$ (horizontal axis) and $q$ (vertical axis). White baseline reflects the null case from the analytic sum without any network inclusion.
(TIF)

**S2 Fig. Approximate model predictions for repulsive interactions between mtDNAs in the network.** $V'(h)$ (left column) and $V(N)$ (right column) under an approximate model for active spread of mtDNA in the network with a radius of $l = 0.1$. Qualitatively trends in behaviour are

captured, but the magnitudes of the effects involved differ from simulated results (see text).
(TIF)

**S3 Fig. Lower-than-binomial variances under more limited mtDNA self-avoidance.** Simulated normalised heteroplasmy variance $V'(h)$ (left column) and copy number variance $V(N)$ (right column) for mtDNAs with mutual repulsion with radius $l = 0.05$ (compared to $l = 0.1$ in the main text) within the network, under symmetric cell divisions. Rows show different values of initial mutant proportion, with $h = 0.1$ in the top row and $h = 0.5$ in the bottom row. The three columns for each panel give decreasing network heterogeneity, expressed via different seed numbers, 4, 16 and 64 (more seed points give a more homogeneous network). In each panel, variances are given for different values of wild and mutant type network inclusion parameters $p$ (horizontal axis) and $q$ (vertical axis). White baseline reflects the null case from the analytic sum without any network inclusion.
(TIF)

**S4 Fig. Comparison of simulation results with analytic model results for first and second order Taylor expansion.** By row, simulation, first, and second order analytic results for $V'(h)$ for random placement of mtDNAs in the network. The first order theory in the second row produces results similar to our simulation results on the off-diagonal, but fails to reproduce the increase observed along the diagonal. The second order theory, while loosely retaining the same structure on the off-diagonal as the first order theory, overestimates this increase along the diagonal. We expect that our model would captures this behavior if we were to derive even higher order terms.
(TIF)

**S5 Fig. Comparisons of individual simulation moments (vertical axes) and analytic moments (horizontal axes) for random spread of mtDNA in the network (Eq 9) for varying proportions of cell volume inherited by the smallest daughter.** Colors reflect the proportion of parent cell volume apportioned to the daughter of interest.
(TIF)

**S6 Fig. Comparisons of individual simulation moments (vertical axes) and analytic moments (horizontal axes) for active spread of mtDNA in the network (Eq 8) for varying parent cell cytoplasm proportions inherited by the smallest daughter.** Colors reflect the proportion of parent cell volume apportioned to the daughter of interest. The repulsive radius is $l = 0.1$.
(TIF)

## Author Contributions

**Conceptualization:** Robert C. Glastad, Iain G. Johnston.

**Formal analysis:** Robert C. Glastad, Iain G. Johnston.

**Funding acquisition:** Iain G. Johnston.

**Investigation:** Robert C. Glastad, Iain G. Johnston.

**Methodology:** Robert C. Glastad, Iain G. Johnston.

**Project administration:** Iain G. Johnston.

**Software:** Robert C. Glastad, Iain G. Johnston.

**Supervision:** Iain G. Johnston.

**Visualization:** Robert C. Glastad, Iain G. Johnston.

**Writing – original draft:** Robert C. Glastad, Iain G. Johnston.

**Writing – review & editing:** Robert C. Glastad, Iain G. Johnston.

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
