## [Decision Letter · Decision Letter 0]

6 Nov 2022

Dear Dr Johnston,

Thank you very much for submitting your manuscript "Mitochondrial network structure controls cell-to-cell mtDNA variability generated by cell divisions" for consideration at PLOS Computational Biology.

As with all papers reviewed by the journal, your manuscript was reviewed by members of the editorial board and by several independent reviewers. In light of the reviews (below this email), we would like to invite the resubmission of a revised version that takes into account the reviewers' comments.

We cannot make any decision about publication until we have seen the revised manuscript and your response to the reviewers' comments. Your revised manuscript is also likely to be sent to reviewers for further evaluation.

Sincerely,

Stefan Klumpp

Academic Editor

PLOS Computational Biology

Daniel Beard

Section Editor

PLOS Computational Biology

Reviewer's Responses to Questions

**Comments to the Authors:**

Reviewer #1: The paper by Glastad and Johnston reports the results of a simulation-based study seeking to understand how mitochondrial network structure controls cell-to-cell mtDNA variability generated by cell divisions. The study includes both a spatial model where random networks are generated using a simple algorithm and also a statistical (non-spatial) model. The study concludes that asymmetric cell divisions induce more mtDNA variability and that MtDNA self-avoidance can reduce heteroplasmy variance. I enjoyed reading this work. I found the study innovative since to my knowledge there has not been much simulation work done on mtDNA variability, and even less using an explicitly spatial model. However some of the main features of the non-spatial model, e.g. asymmetrical division, have appeared before in models of partitioning of mRNA and proteins during cell division. As well its unclear to me how the details of the cell size regulatory mechanism influences the fluctuations they are studying and I think it would be important to extend the model to study such effects. My more detailed questions are below:

Major

(i) In recent years, asymmetric cell division has been studied by other authors. For e.g. a model investigating the role of asymmetric cell division on fluctuations in mRNA and protein numbers was studied in Jia & Grima, "Frequency domain analysis of fluctuations of mRNA and protein copy numbers within a cell lineage: theory and experimental validation." Physical Review X 11.2 (2021): 021032. How the degree of asymmetry affects the cell size distribution was also studied in Jia et al. "Cell size distribution of lineage data: analytic results and parameter inference." iScience 24.3 (2021): 102220. These references and similar should be discussed in the Introduction or elsewhere.

(ii) Its unclear to me how the variability in the cell division time would affect the results presented. Previous studies (see papers above and also "Concentration fluctuations in growing and dividing cells: Insights into the emergence of concentration homeostasis" https://doi.org/10.1371/journal.pcbi.1010574) have shown that this signficantly affects the noise in newborn daughter cells when coupled with binomial partitioning at cell division. The variability in the cell division time is intimately linked to the strategy (timer / adder / sizer) by which cells regulate their size which will naturally lead to variations in the cell size at division and so I presume that hence these strategies must influence the fluctuations in mtDNA. This could be addressed by extending the simulations to develop a spatio-temporal model of the whole cell cycle, from birth to division, capturing the mtDNA variability over an arbitrary number of generations. If this is difficult then I suggest to at least address via discussion.

Minor

(i) its unclear how well the properties of real mitochondrial networks are approximated by the simulation algorithm. The authors mention that it is not intended that these are matched perfectly well. However a more precise statement on the agreement of certain statistical properties of the simulation and real networks would be useful.

(ii) Line 104, the quantity V(h) is undefined at this point in the main text.

Reviewer #2: This paper carefully considers, through a mix of computation and analytics, how selective network fusion, network spatial heterogeneity and division have a bearing on the fluctuations in mitochondrial copy-number and heteroplasmy through division. The close alignment of analytic and computational effort ensures that this is a good contribution to the literature. There are some non-trivial results, for example showing that having a homogenous network with selective network fusion can decrease heteroplasmy variance. I think there are some aspects of flow/clarity that could be improved. In part I suspect that this issue is induced by fragmentation of the content into main, methods and supplement.

Comments:

“the dimensionality of the model does not affect the statistical considerations of partitioning” why not?

There are issues with logical flow that start somewhere around ’Statistical models of mtDNA inheritance…’ line 115. There is a missing logical step -- or I missed it -- that having a heterogeneous network but mtDNA somewhat uniformly distributed across it will naturally create variance in copy-number between the halves: one half of the cell could naturally have more network than the other half. These large-scale fluctuations will not be present if we remove the spatial structure. This lack of clarity is probably induced by the fact that the detail of the cell division mechanism is not clarified before the reader is introduced to the results in figure 2. In fact in general the cell division mechanism isn’t that clearly articulated with some peculiar wording about \\phi.

The connection to reference 53 could be strengthened outlining their computational network models and differences from their work.

It may be helpful to some people to point out that the model at line 123 is a hierarchical model where you make the specific choice not to model w_i and m_i as RVs but instead use expected values (I don't know whether explicitly treating them in RVs would have implications for the observed variance in W_i and M_i).

The scale l for repulsion is insufficiently motivated. Clarify the size of l compared to other scales in the system. Does changing its relative scale have an effect? I’m not requiring that the text include further simulations but some indication about its expected role.

Line 210 ‘network inclusion discrimination lowers the heteroplasmy variance induced by cell divisions’ since this is one of the more unexpected results more intuition should be given on why this occurs.

Line 215 I’ve misunderstood something: why is this surprising? -- even with exactly the same number of mtDNAs in both partitioned parts there can be fluctuations in their heteroplasmies.

For replicating cells it seems at least credible, especially if a subset of the mtDNA are the only ones replicating, that the network will have sets of the, genetically identical, daughters of source nucleoids close to each other within the network. In principle this could increase heteroplasmy variance on division. The trade off between the timescales of mt-dynamics/nucleoid-diffusion mixing (and degree of fragmentation/diffusion on division) and mtDNA replication would scale this effect. With a limited number of nucleoids being able to replicate this effect could possibly be significant. Perhaps worth alluding to this?

I wasn’t convinced of the value-add of the diffusion section — perhaps requested by an earlier reviewer. If the particular role of this section could be clarified that would help — but not vital (perhaps it is reassuring in that nothing surprising happens). What should be clarified is how the diffusion occurs — presumably the network structure gives the ic’s but is thereafter irrelevant. The confounding alternatives would be diffusion of nucleoids within the network or diffusion of the organelles.

Figure 6 is a very helpful one -- you could mention it at the end of the introduction so that readers have in mind the key results as they proceed.

In Figure 6 I didn't quite understand why, for the homogenous network, the introduction of genetic bias alone should serve to slightly increase copy-number variance relative to its absence. I.e. in the bottom row why aren't both red curves '- -'. Doubtless an error on my part but wherever this is covered in the text should possibly be expanded.

Line 300 any justification for these parameters?

I have not carefully checked the supplement -- while the methods seem sensible my silence on this point is not a guarantee of them being free of errors.

Typos etc:

Consistent italicisation of s.

Around line 172 there is some unclear wording with the phrase ‘challenge’

Axis numbering is pretty small in the heat map figures.

Odd use of the word facet — clarify.

Line 254 ‘as observed experimentally and computationally in…’

In Fig 6 active and random spread are used — but they aren’t sufficiently used in the rest of the ms.

I repeatedly read the top row of figure 6 as copy-number -- maybe because it's a simpler quantity than heteroplasmy. You could flip the order?

Some equations in the supplement should be spread over two lines

Line 317 ‘every point of the network was equally like to host’

**Have the authors made all data and (if applicable) computational code underlying the findings in their manuscript fully available?**

Reviewer #1: Yes

Reviewer #2: **No: **I missed a link to a git hub for the code -- it might have been provided -- having such a link would be useful though admittedly the computational models are simple.

PLOS authors have the option to publish the peer review history of their article (what does this mean?). If published, this will include your full peer review and any attached files.

Reviewer #1: No

Reviewer #2: No
---

## [Decision Letter · Decision Letter 1]

15 Feb 2023

Dear Dr Johnston,

We are pleased to inform you that your manuscript 'Mitochondrial network structure controls cell-to-cell mtDNA variability generated by cell divisions' has been provisionally accepted for publication in PLOS Computational Biology.

Best regards,

Stefan Klumpp

Academic Editor

PLOS Computational Biology

Daniel Beard

Section Editor

PLOS Computational Biology

Reviewer's Responses to Questions

**Comments to the Authors:**

Reviewer #1: The authors have replied to my questions and made suitable changes. This is a great paper and I strongly support its publication in PCB.

**Have the authors made all data and (if applicable) computational code underlying the findings in their manuscript fully available?**

Reviewer #1: Yes

PLOS authors have the option to publish the peer review history of their article (what does this mean?). If published, this will include your full peer review and any attached files.

Reviewer #1: No

---

## [Editor Report · Acceptance letter]

18 Mar 2023

PCOMPBIOL-D-22-00964R1 

Mitochondrial network structure controls cell-to-cell mtDNA variability generated by cell divisions

Dear Dr Johnston,

I am pleased to inform you that your manuscript has been formally accepted for publication in PLOS Computational Biology. Your manuscript is now with our production department and you will be notified of the publication date in due course.

With kind regards,

Zsofia Freund
